# Investigation of Spatial and Cultural Features in Contemporary Qatari Housing

Asmaa Al-Mohannadi [1], Mark David Major [1], Raffaello Furlan [1], Rashid Saad Al-Matwi [1] and Rima J. Isaifan [2,*]

1 Department of Architecture and Urban Planning, College of Engineering, Qatar University, Doha P.O. Box 2713, Qatar
2 Division of Sustainable Development (DSD), College of Science and Engineering, Hamad Bin Khalifa University/Qatar Foundation (QF), Education City, Doha, Qatar
* Correspondence: risaifan@hbku.edu.qa

**Abstract:** Housing is a basic human need and a fundamental component of settlement status. The architectural form and spatial provisions of housing evolve in line with—and transform to meet—a specific era's needs. Globalization has been responsible for changing the nature of housing in Qatar over the last thirty years. It has led to a standardization of construction methods and built form, representing a dramatic departure from past models of vernacular residential architecture. In light of these challenges, the ultimate purpose of this study is to explore the spatial and cultural features in a small sample of contemporary housing in Qatar. It explores the spatial layout of four Qatari residential villas to assess the social and cultural roles in contemporary housing models against the background of previous research. In the study, the authors utilized space syntax as an analytical tool to demonstrate patterns of visibility and room relations in the samples to understand occupants' system of activities in the contemporary domestic setting, deploying visibility graph analysis (VGA) and relational graphs. Key findings include the interpretation of the probable relation to socio-cultural factors such as gender roles, hospitality, and privacy. Hence, this study fills gaps in knowledge about Qatari and Middle Eastern housing today.

**Keywords:** spatial analysis; domestic space; housing; visibility; space syntax; Qatar

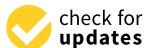



## 1. Introduction

Housing is a fundamental part of the city's urban morphology. It responds to the demographic and economic realities that might alter and transform urbanism in physical and socio-cultural terms. Housing provides a key indicator of urban transformations over time. It is also a means to assess the socio-cultural pillars of sustainability because the house form embodies and reflects specific people's cultural and social practices in everyday life. In Qatar, as in the broader Gulf Cooperation Council (GCC) region, globalization and rapid urbanization have affected residential architecture in several ways. Scholars attribute these changes to the standardization of building practice, construction methods, and urban planning regulations tied to the influence of global trends in managing mega-scale architectural projects [1–3]. These trends deviated from previous trends for the vernacular formation of residential housing with distinguishing characteristics embedded in architectural identity and socio-cultural background.

This study seeks to define social and cultural roles at work in the spatial form of contemporary housing models in Qatar. The focus is the multi-story residential villa, the most widespread housing model during an era of rapid urbanization and globalization in Qatar at present [4]. The four sample houses of the study are contemporary residential villas in the suburbs of Doha and Wakrah cities. The paper utilizes space syntax to examine the role of socio-cultural factors in the modern house form. Much material is available in the literature about space syntax and its domestic, building, and urban applications [5,6].

We used the current state of knowledge in architecture and sociology in coordination with space syntax analysis to enrich concepts of the architectural identity in Qatari housing as places to dwell for the people who live there. The research investigates how housing and society interrelate in domestic space and what house form can tell us about its social context. At the core of the paper is a simple question. How are social and cultural roles embedded in the spatial form of contemporary Qatari residential villas as determined by the system of inhabitants' activities based on room functions?

These objectives aim to answer four main research questions. First, is it possible to infer vernacular housing models as a foundational source of architectural identity in these houses with reference to previous research about traditional Qatari house forms, allowing for comparative analysis of contemporary housing as a dependent variable against vernacular housing as an independent variable? Second, it is possible to enrich our definitions of the social sustainability of architecture by an interdisciplinary study of society's socio-cultural attributes as embedded in its housing, thus linking this to the architectural design process itself. Third, it is possible to understand better the nature of urban transformations arising from contemporary trends in the spatial form of houses against the background of our current state of knowledge in the field and, finally, using space syntax tools to assess the effectiveness of computer simulations of contemporary residential villas in architectural form, spatial distribution, and domestic life. The first and second are aspirational goals of the research, whereas the third and fourth target more realistic, short-term research outcomes.

The need for qualitative and quantitative analysis of these socio-cultural aspects of house form led to using space syntax as the primary methodological tool of the study. By using computer modeling, the goal was to simulate the enduring relationship between house form and culture. It was expected that the spatial form of these contemporary houses might lack the essence of the socio-cultural values of Qatari society, and their spatial layout might be insensitive to these values. Hence, this study was initiated anticipating that the standardization of construction methods and built form and the formation of a consumption culture based on Westernization would lead to an absence of reference to traditional Qatari vernacular architecture of the past. The results of this work show that this is not precisely the case. The picture that emerges about contemporary Qatari housing is more nuanced.

## 2. Background

This section reviewed the literature and divided it into three themes to briefly examine the current state of knowledge in the field. The first theme summarizes the disciplinary context, specifically housing and domestic space research on the cultural roles affecting house form. The second theme focuses on housing dynamics in the contemporary era across the Middle East and North Africa (MENA) and the Arabian Gulf regions. The regional examination's purpose is due to the limited references available in the literature about Qatari residential architecture. Hence, relevant studies addressing housing typography or topics concerning culture in domestic architecture in the region were highlighted. A key takeaway is a need for previous studies using space syntax in the GCC region. The third theme briefly reviews the use of space syntax theory and its application in domestic settings, demonstrating the novelty of the current study for analyzing the spatial culture of contemporary Qatari housing models.

The disciplinary background focuses on the relationship between house form and culture, which is the critical theme of Amos Rapoport's' academic publications and books [7–9]. Rapoport is a well-known scholar in the field of architecture and environment-behavior studies. He establishes a theoretical framework about aspects of culture, human behavior, and architectural form in the evolution of housing and other building types. His framework, including this study, has formed the foundation for subsequent studies about housing worldwide over the last three decades. It includes the significance of the house as a social unit of space as defined by tradition, activity, and public agreement. Rapoport and Kent describe the house's spatial arrangement as a matter of the occupants' lifestyle activities

and cultural aspirations [10]. They argue that the system of activities reflects domestic space based on six components. These components address elements of the time (the question of when), occupants or actors (the question of who), the activity (the questions of what, how, and why), and the environmental setting (the question of where). These have consequences on the nature of activities and the persons involved, such as gender distinctions, location, privacy, specifics related to the epoch, the inter-relationship of different activities, and their meaning in the culture [11].

The subsequent research about house form and culture has resulted in multiple theories and applications, facilitating the connection between two concepts (physical and social) in housing research [12,13]. It has led to a deeper understanding of the role of the built environment in human societies and cultures worldwide. Among the most important international research programs examining housing over the years has been space syntax. In the late 1970s and early 1980s, Hillier, Hanson, and many others in The Bartlett at University College London (UCL) envisioned new ways to represent and measure architectural and urban space that took place in the innovative methodologies of space syntax. It led to many theoretical contributions to our knowledge about the sociology of the built environment [14–16]. Space syntax representations account for generic human movement, occupation, and visibility activities in a built environment. At a basic level, this includes geometrical concepts of a point as a static position in space, the axial lines as a component of linear movement through space, and a convex space where all points are visible to each other. Over four decades, researchers worldwide have built a considerable body of findings on the built environment, from houses to entire cities, using space syntax in published books, journal articles, and conference papers such as the biennial International Space Syntax Symposia proceedings [17–23].

Space syntax translates its methodologies and measurements into user-friendly software programs enabling researchers to model plans at a building or urban scale and perform tests such as design alterations. One of the most useful is visibility graph analysis (VGA). It represents the matrix of visual fields from a gridded set of points in a closed system, i.e., closing exterior doors to the outside world. It can be beneficial for noting aspects of culture in house form, such as privacy in segregating spaces [19,21,24]. There are multiple metrics and spatial measures available in VGA modeling. However, for this paper, we focused on connectivity, which measures the primary connection of an individual space to adjacent spaces, and integration, to better represent the shallowness or deepness of a space in terms of mean depth [25]. This paper's research design section provides further definitions and clarifications about the software and the selected tests for the case study houses.

Many domestic architecture studies in the MENA region and the Arabian Gulf examine socio-cultural factors affecting house form and spatial layout. Elmansuri and Goodchild explored the sociospatial components of Arab homes in response to issues of tradition, modernity, and gender in the Libyan context [26]. They found that religious and cultural interpretations of privacy and gender segregation played a role in defining the home's meaning for residents. Their findings have implications for understanding Arab domestic space in traditional courtyard dwellings and contemporary housing schemes. In Libya, Emhemed and Ujam tested the applicability of space syntax tools for generating housing regulations in the retention of traditional architecture [27]. Unlike the current study, their research incorporated building and urban-level analysis to better understand domestic spatial layouts' internal functioning and the dwelling's relationship to the external urban setting. Based on their findings, the researchers proposed building and planning regulations to address spatial integration (or relativized mean depth, a significant mathematical conception in space syntax analysis), social interaction, social behavior, land uses, and segregation and safety.

Giddings, Almehraj, and Cresciani examined housing issues in the central region of the Arabian Peninsula [28]. The study shed light on the transition of architectural trends in the area. Due to rapid economic growth in the 1950s, residents abandoned the culturally responsive traditional courtyard house in favor of the detached residential villa, which

became the prominent housing model for city dwellers. It is an oft-repeated trend elsewhere in other countries of the Arab Gulf region, including Qatar. By using questionnaires, surveys, and building analysis, the researchers confirmed that several families found it challenging to re-embrace the domestic lifestyle associated with the traditional courtyard house model due to changes in economic status, climatic discomfort, space limitations, and its association with a less prosperous era. On the other hand, they perceived the lack of privacy and over-reliance on air conditioning for micro-climate enhancement in the detached residential villa as a significant disadvantage of the contemporary housing model. The researchers' proposed solution aimed at developing new housing schemes to preserve residents' social and cultural values while more effectively responding to the technological opportunities available for contemporary housing. Almehrej later contributed proposals for designing contemporary homes on the Arabian Peninsula on this basis [29]. He developed a comprehensive design guide to meet the challenges of spatial form in modern housing. The design guide covers three primary principles: (1) human needs, such as climatic comfort and privacy; (2) place needs, such as attachment and identity; and (3) house form needs, such as the type/nature of spaces and other technical specifications.

Three significant studies have provided benchmarks for studying socio-cultural patterns in the spatial layout of Qatari housing. The first is the contribution of Sharon Nagy, an American anthropologist working in the Gulf region during the mid-1990s [30–32]. She analyzes the appearance and form of contemporary Qatari housing and the (re)production of social relations due to the transformative forces of globalization and rapid urbanization. Nagy focused explicitly on the majlis space as an aspect of domestic hospitality in Qatari dwellings, using this space as a social marker for transformations of house form over time. In domestic architecture, a majlis is a common term in Arabic and Persian (meaning "council"), referring to a private space to receive and entertain guests associated with hospitality, equivalent to a lounge or salon in British English and French, respectively. Based on this, she outlined the chronology of changes in Qatari house form over three phases: (1) the yard-oriented house in the pre-1960s; (2) the public house in the 1960s–1970s; and (3) the senior staff villa house in the 1984s–1990s [30]. Nagy also traced the history of urbanism in Qatar from the earliest vernacular settlements and reviewed geopolitical changes influencing significant economic progress due to oil production and exportation. She argued that this has led to a massive transformation in the Qatari-built environment with widespread implications for domestic relations and wealth distribution. Nagy also sheds light on Qatari preferences for housing location and composition, which indicates the influence of greater social diversity on the residential patterns of settlement form.

Sobh and Belk reported the results of a qualitative study of Qatari houses in the early 2000s [33,34]. They found that Qatari households retain a keen sense of domestic identity by maintaining privacy and gender segregation as cultural anchors in response to the perceived shortcomings of economic globalization, rapid urbanization, and construction standardization. Nonetheless, they argued for society to reconsider the ongoing transformation of house form as an aspect of national identity due to the socio-cultural significance of housing, economic wealth and citizenship privileges, and demographics in Qatari society. They identified the predominant need for (1) privacy in contemporary housing models and (2) detailed physical and social analysis to understand better female-dominated spaces in the domestic setting.

More recently, Wiedmann et al. examined spatial fragmentation as an aspect of Qatar's new housing patterns utilizing space syntax tools and GIS assessment on a citywide scale [35]. Their study identified three main developments for housing in the capital city of Doha from 2004 to 2017. It includes (1) high-density zones transitioning from old Doha to suburban areas, (2) residential options for high-income residents in luxurious waterfront developments, and (3) gated suburban compounds. It built on previous studies about the sequence of changes in domestic typologies in the Gulf Region, charting the nature of housing transformations related to social, economic, and political changes. It reviewed housing typologies utilizing a sample of housing plans in Gulf cities, outlining historical

phases for the evolution of housing from (1) post-nomadic, (2) traditional, and (3) modern to (4) contemporary since the 1990s [2,36]. The latter is essential for the research in this paper. It forms the basis for defining contemporary housing models based on international standards of construction and allocation due to increasing land values, a liberalized housing market, and a limited supply of public housing. Wiedmann et al. argued for a comprehensive planning and design process, which places contemporary residential architecture at the center of a strategy for sustainable urbanism. They concluded that delivering equitable and affordable housing options should form the pillars for socio-cultural and environmental sustainability in the country [37,38].

This area is where space syntax could prove valuable. Its theories and methodologies allow architectural and urban researchers to examine socio-cultural relations within a domestic setting and outward to the larger urban context to mitigate between different scales of the built environment [14–16]. In their study of the spatial reasoning in the programming of socio-spatial parametric grammar, Al-Jokhadar, and Jabi utilized the state-of-the-art computational models of shape grammar and syntactical analysis to explore the social dynamics of vernacular houses across the MENA region [39]. For their use of space syntax analysis to investigate domestic culture, Al-Jokhadar and Jabi deployed three different software packages: (1) Agraph, (2) sytax2D, and (3) DepthMapX. The last is relevant to the current study. We utilized the visibility graph analysis (VGA) tools of DepthMapX to represent connectivity and integration to understanding better the social construction of space in contemporary Qatari residential villas [24,40]. Their study provides a rich source of updated methodological techniques for analyzing residential dwellings.

Alitajer and Nojoumi also examined home privacy concepts using space syntax [41]. They conducted a comparative analysis of the transformation of socio-spatial parameters for residential architecture in Iran [42]. They argued that privacy is a fundamental human necessity due to socio-cultural factors, which play an essential role in developing residential architecture in Muslim regions. They demonstrated that much lower connectivity levels characterize the entry areas to traditional Iranian houses than modern Iranian ones. Alitajer and Nojoumi argued that this challenges privacy in Muslim homes. The design of entryways plays a crucial role in generating visual and functional privacy by subtly separating the house's public and private domains. It is an essential consideration for maintaining the sanctity of the household in culturally conservative societies, which contemporary housing models often lack in their design.

Khattab adopted a similar approach to study the social construction of space in traditional Kuwaiti houses [43]. He followed the standard methodological approach in space syntax based on defining spaces of the domestic setting, spatial representation using permeability graphs and axial maps, and analysis of the spatial network based on syntactic measurements. He utilized the research findings for the sample of seven Kuwaiti courtyard houses to suggest a standard model for the housing typology in documenting the heritage of historic residential architecture in Kuwait. Similarly, socio-spatial analysis plays a vital role in defining the analytical parameters for documenting vernacular architecture in the region as part of the recent "Gulf Sustainable Urbanism" research endeavor of Harvard University, sponsored by the Qatar Foundation [44]. The socio-spatial component of this research includes integration, maximum visual step depth (the maximum number of steps necessary to transverse from any location to any other based on the pattern of visibility fields available in the built environment), and justified graphs based on permeability (the state of allowing to pass through from one space to the next). Illustration and model simulation of layouts supported the research efforts.

There has been a great deal of house form and culture research since the 1960s and 1970s, deepening our knowledge about the relationship between spatial layout and human behavior in domestic settings, including space syntax research, more so before the dawn of the 21st century. However, over the last decade, examples of such literature have been strangely lacking, especially in Western societies. A brief sampling of the recent literature reveals only studies about the psychological effects of overcrowding on children

in domestic settings [45], behavioral factors involved in domestic water/energy usage and conservation [46–50], the relationship between mental health and housing quality in slum rehabilitation [51], nuclear bomb shelters as a rite de passage for children in Swiss domestic settings during the 20th century [52], and ascertaining visual comfort in off-grid homes [53].

Only a small amount of the literature bears any resemblance to traditional research about domestic form and culture, including examinations of room space size about intimate activities [54], experiments about the perception and conception of a sample of house floor plans [55], understanding the implications for domestic lifestyles in non-urban locales [56], examining socio-biological control in troubled families in the UK [57], the philosophy of marking and claiming by people dwelling in domestic residences [58,59], and the impact of modern communication devices on domestic lifestyles and design in Australia [60]. It is difficult to ascertain the reasons for this relative lack of interest in the literature. However, it may be related to decreasing household sizes in Western societies post-war. Our literature review demonstrates great interest in house form and domestic lifestyles in regions such as MENA, where the average household size is still large. In many ways, this indicates a significant need to reignite the conversation about house form and culture first began by Amos Rapoport over a half-century ago.

The study in this paper adopts such an approach to assess the socio-spatial nature of contemporary Qatari houses in a small sample to unveil their domestic culture. It builds on previous findings of the socio-spatial analysis of housing in the Arabian Gulf and the larger MENA region. By definition, such research tends to be multidisciplinary, incorporating aspects of architecture, urban design and planning, sociology, ethnography, and anthropology to one degree to another. There is a need for such housing research as concepts of sustainability gain increasing importance worldwide and in the State of Qatar for the long-term benefit of the people who live in our built environments.

The paper hypothesizes that the spatial form of the houses should embed cultural and social attributes drawing on Rapoport's conception of the house as a social unit of space [7]. Using Rapoport's concept, the study proposes that the spatial structure of the houses should integrate cultural and social features. For our purposes, the dependent variable is the spatial form of the house. The independent variable is the socio-cultural factors at work. The latter becomes embedded in the spatial layout of the former. The ground floor layout of these houses is given special attention in the article. Due to the small size of the dwellings in the sample, the range of room functions on the ground floor allows us to examine gender roles, hospitality, and privacy adequately. In these residential villas, the main suite of family bedrooms produces seclusion through simple separation on different floors in sections.

## 3. Methods and Research Design

We developed a methodology for the research to evaluate the contemporary residential villas based on three parameters or criteria, namely the (1) spatial form of the villa within its compound walls, (2) probable activities of the inhabitants based on room functions, and (3) possible cultural roles for the spatial layout. These three parameters define the primary basis for the analysis and subsequent discussions (Figure 1). However, it resides within the larger context of the theoretical framework based on the literature review.

As part of the data collection in this study, we documented general architectural knowledge and oral data to provide additional background to the research. It included several local seminars and events about housing and urbanism in Qatar, such as the virtual launch of the Gulf Sustainable Urbanism Encyclopedia by Msheireb Properties and Harvard University and the Gulf Architecture Project (GAP) by Qatar National Library in collaboration with Liverpool University and Qatar University [44,61,62]. These events and dialogues featuring national pioneers and practitioners in architecture and residential design provided a rich source of information about the urban history and evolution of

housing in Doha and the challenges of urbanism in a rapidly urbanizing country [63]. We refer to this background information as necessary.

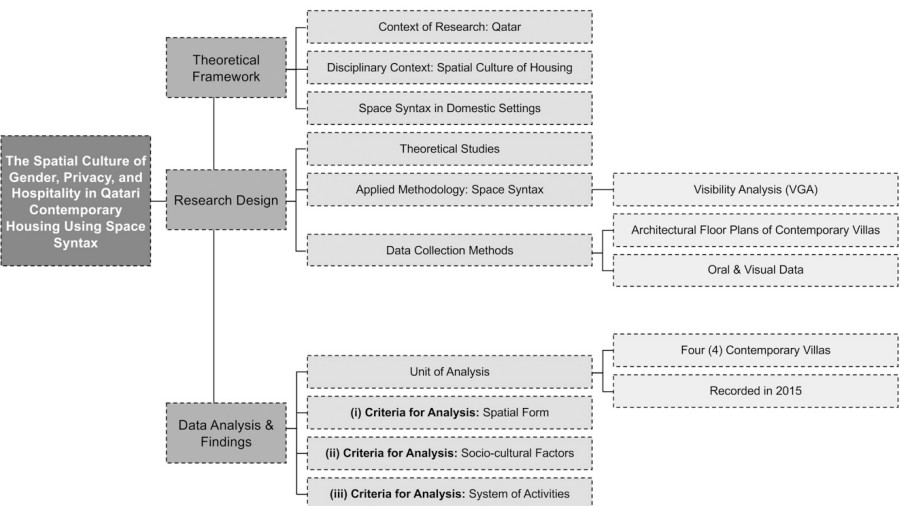

**Figure 1.** The methodological basis for this study (Source: Authors).

In the data collection phase, we selected four residential villas based on various criteria, including the availability of complete plans recorded in the Public Housing Inventory of the State of Qatar, the date of construction circa 2015, and the inclusion of hospitality spaces. Two of the most important criteria were choosing similar plot areas to enable a more straightforward comparative analysis of the spatial layouts and selecting a middle-class Qatari family currently owning the residential villas. Collectively, this helped to foster a socio-spatial analysis of contemporary house forms. The contemporary residential villas' ground floor plans were then redrawn for visual illustration and VGA modeling purposes and set to a common scale (Figure 2).

In terms of the methods of data analysis, the spatial form of the villa considers essential architectural elements (such as columns and stairwells) and configurational analysis of the interior and exterior spaces on the ground floor within building elements such as solids and voids, compound walls, building envelope, articulated entryways, room functions, and other physical aspects. Based on room usage, public/private use areas and the timings of use based on the functionality of the space were defined. Moreover, the social construction of space in the residential villas was analyzed using relational (or unjustified) permeability graphs and VGA to simulate each house's visual integration and connectivity pattern. Then, the relative degree of integration or segregation was reviewed to understand the nature of privacy for each space and gendered and hospitality spaces at the ground level of the house, especially regarding the entry sequence from the street. The VGA tools in DepthMapX color spaces range from red (integrated) to orange, yellow, green, and blue to dark blue (segregated) based on the visual fields available from a gridded set of points relative to their level of connectivity. The evaluation of the system of activities within the household was based on labeling room functions concerning the larger literature available about the domestic culture of Qatari families. This helped illuminate the embedment of cultural roles within the house form. All four villas have a main house and a separate kitchen/servant block. The designs utilize the maximum metric distance available on the plot to locate the kitchen/servant house block as far away as possible from the main gate entrance to the compound. In all four cases, the separate kitchen/servant utilizes (1) a rectilinear shape with (2) a zero-lot line setback in two dimensions of the plot and (3) running along approximately 50% to 60% of the linear length in one dimension. All four residential compounds locate the main house on the lot to impede any views of the kitchen/servant block from the main gate entrance. This block is viewable from a vehicular

gate entry for Villa #1; the blank outer wall is a storage space only accessible from within the kitchen/servant block.

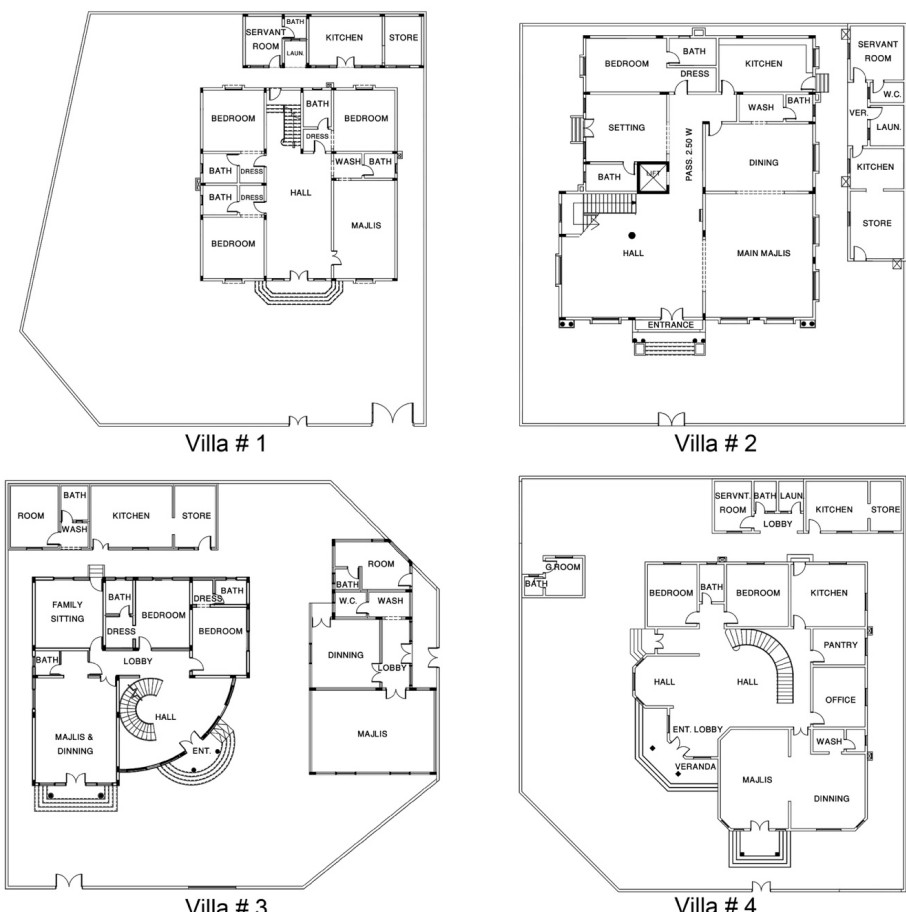

**Figure 2.** The ground floor plan with labeled room functions of the four contemporary villas (Source: Authors).

The finished floor of the main house in all four residential villas elevates about 0.6–0.7 m above the ground level of the exterior yard. The finished floor of all other separate house blocks in all the residential villas is the same (or nearly) as the outer yard ground level. Villa #3 possesses a third separate house block for the male majlis, dining, and guest room suite with its own entry from the street. It is the only villa with separate entries on different faces of the plot. Villa #4 also possesses a small, separated block for the gardener's quarters. The main entry gate from the street for the two villas aligns on an axis with the main entry of the main axis. The exceptions are Villa #3 and #4, where the principal entry to the main house offsets at a 45° angle about the gate entry of the compound from the street. Both gate entries marginally shift away from an on-axis alignment with the entry to the female majlis (in Villa #3) or male majlis (in Villa #4) space of the main house. In both cases, it is only due to the more extended width of the compound entry gates since the alignment of the hinges for the swing of one of the gates/doors is the same. However, interestingly, the additional threshold width for the compound entry gate occurs in the direction away from the main house entry in both cases.

Villa #1 has a main house with the smallest building footprint in the sample and the largest exterior yard space in the compound. The building footprint sets back in the lot skewed to one corner with a separate kitchen and servant block running along the property line. It has two entries from the street to the compound: a larger one for vehicles and the main footpath entry on the axis with the villa's front door. The vehicular entryway indicates why more exterior yard space is available in Villa #1 compared to the other residential

villas, i.e., parked vehicle storage. Villa #2 is the only house plot that is rectangular in shape, only 4.5% off in one dimension from being a perfect square. Villa #4 would be only 7.5% off in one size from being a perfectly square shape, except for its one 45° chamfered corner. The same applies to Villa #2, except for its two 45° chamfered corners. Villa #1 would sit within a perfectly square shape, except for its chamfered corners at acute angles. The lot area coverage of Villa #1 is about 35%, with the main house footprint accounting for over three-and-a-half times that of the separate kitchen/servant block. The lot coverage of Villa #2 is about 55%, with the main house footprint accounting for over five times larger than the separate kitchen/servant block. For Villa #3, the lot coverage is about 45%, with the main house footprint accounting for over two-and-a-half times larger than the separate kitchen/servant block. The footprint of the majlis/guest suite block in Villa #3 is about 60% of that of the main house. The lot coverage of Villa #4 is 45%. The building footprint of the main house is over five-and-a-half times larger than the combined kitchen/servant blocks.

The study in this paper presents applied research through the application of space syntax in the contemporary domestic architecture setting. It has the potential to solve a specific problem related to contemporary Qatari housing, namely, to investigate the degree of socio-cultural integration in house form arising from the adoption of standardized construction methods and building code requirements in the State of Qatar in the early 21st century. The methodology of the study satisfies a need for assessing the built environment through computer simulations and graphical representations of modern house forms. Therefore, it allows researchers to the urban transformation of Doha and contemporary trends in the spatial form of housing based on measurable outcomes and reasonable interpretation of results. The study utilizes space syntax techniques and its open-source software program DepthMap05.x. The software models an objective description of spatial layout using architectural drawings of buildings at various levels of detail [13,16,64,65]. The research output involves environmental visualization and simulation utilizing relational graphs and visibility analysis to evaluate any discernible hierarchal arrangement of public and private spaces in the domestic setting, which might represent the embedding of socio-cultural roles in the house form [66].

The process occurred using two procedures for modeling and visibility simulation. First, we simulated the interior spaces of each housing block, excluding all exterior yard spaces. The purpose was to understand the pattern of visibility in the interior spatial layout of each building for distinguishing some areas of family use from hospitality spaces, such as the entry hall and majlis, and other service spaces, such as the kitchen or stairwells. Second, we modeled all interior and exterior spaces collectively within the compound walls to analyze the all-inclusive visibility pattern, including the relationship between the spatial layout of different housing blocks and the exterior yard. We then merged the two different scales of analysis using raster graphics editor software by placing the interior-only visibility analysis output over the top of the all-inclusive one. The purpose was to provide a simulation of each house that simultaneously represents the viewpoint for two different scales of the contemporary house form. The first is the resident's viewpoint of the interior-only spatial network. The second is the visitor's viewpoint of the residential compound, exterior yard, and separate housing blocks. This graphical technique demonstrates the user experience as the scale of their perception of the domestic settings changes from outside to inside or vice versa. This visibility graph analysis overlay provides a more informative collective picture of contemporary residential villas than the separate simulations, i.e., interior-only and all-inclusive. As such, it represents a methodological innovation of space syntax techniques for visibility graph analysis.

## 4. Results

In space syntax terms, a convex map is an outcome of subdividing the architectural plan into defined spaces [67]. We began with defining the convex spaces in the houses (Figure 3), followed by the simple comparison of relational graphs based on convex mapping of the spatial layout in the residential compounds. These represent unjustified

permeability graphs (Figure 4). The graphs illustrate the relationship of spaces to each other based on simple connectivity from one space to the next, with each space represented as a node and any connection between them represented as a line. We color-coded the nodes in the relational graphs for specific room functions. It includes the public street exterior space, entry foyer/hall, living and dining rooms, kitchens, male and female majlis, and bedrooms in the ground-level plan. The graphs indicate vertical transitions with an arrow, except for the elevator in Villa #2. This provides a simple picture of functionality in each residential compound. The more a space connects to other spaces, the more critical it becomes in the spatial layout—the fewer the connections, the more isolated the space within the overall layout [13,16,68,69].

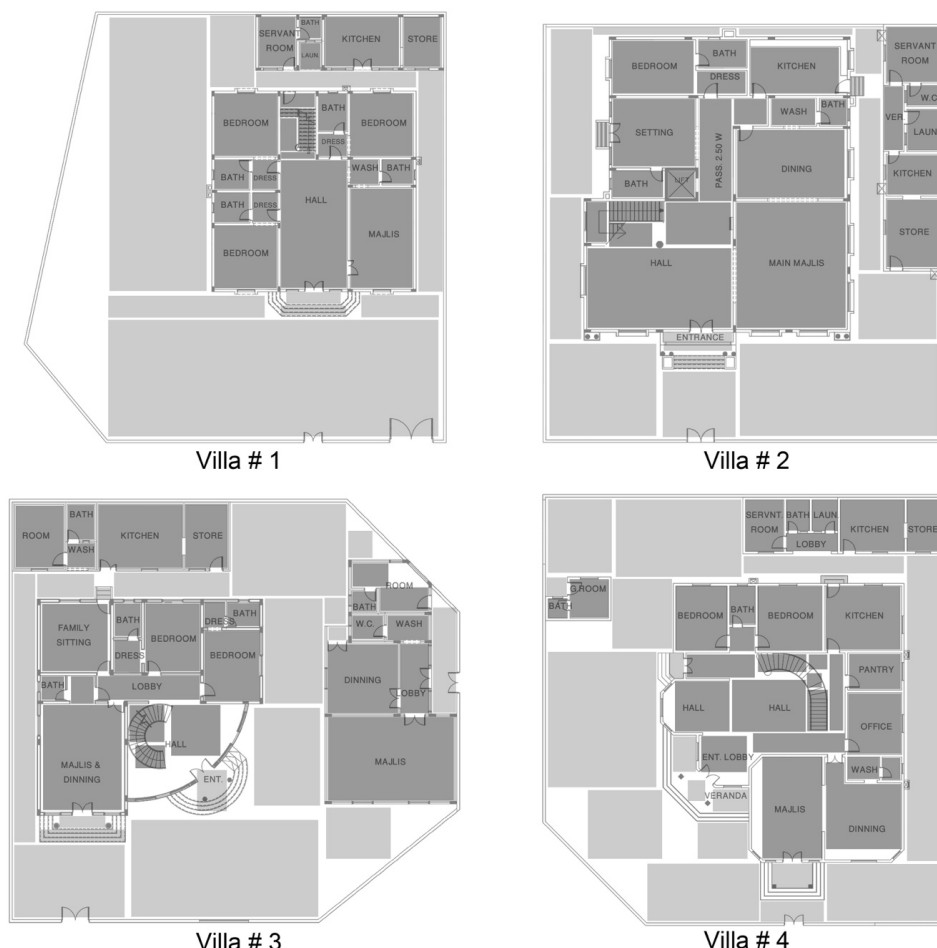

**Figure 3.** Convex spaces for the ground floor plans of the four contemporary villas are dark grey for interior spaces and light grey for exterior spaces (Source: Author).

All four main houses utilize a large entry foyer or hall. Interestingly, the entry hall of the main house for Villa #1 serves as the main distribution point in the layout with seven connections. There are direct connections to six distinct parts of the house and one to the exterior yard via the main entrance steps. In contrast, the entry foyer/hall in the main house of the other three villas is not the main distribution point in the layout, with connections to only 2–3 distinct parts of the house. In all three, the main control points for distribution in the spatial layout occur one additional step further inside the house via articulated hallways, which serve as vestibules to different wings. This is clearest in Villa #3, with the main entry hall connecting to only a vertical transition, hallway vestibule, and the exterior portico for the main entry. The entry hall to the main house of Villa #1 must serve this role as a distribution point due to its smaller size and lot coverage (35%) compared to the other villas (>45%). The villas with the most extensive lot coverage incorporate more

elements of architectural articulation in the entry sequence for visitors, whether involving the entry to the main house or a separate entrance to hospitality spaces. It includes stairs rising to the finished floor elevation, an entry portico with columns, or a transition via intermediate convex spaces in the separate hospitality block of Villa #3. Villa #1 possesses an insignificant localized circulation ring with the exterior street due to the vehicular gate. Villa #3 has a large, complicated ring of circulation with the exterior street due to the separate entry to the hospitality block.

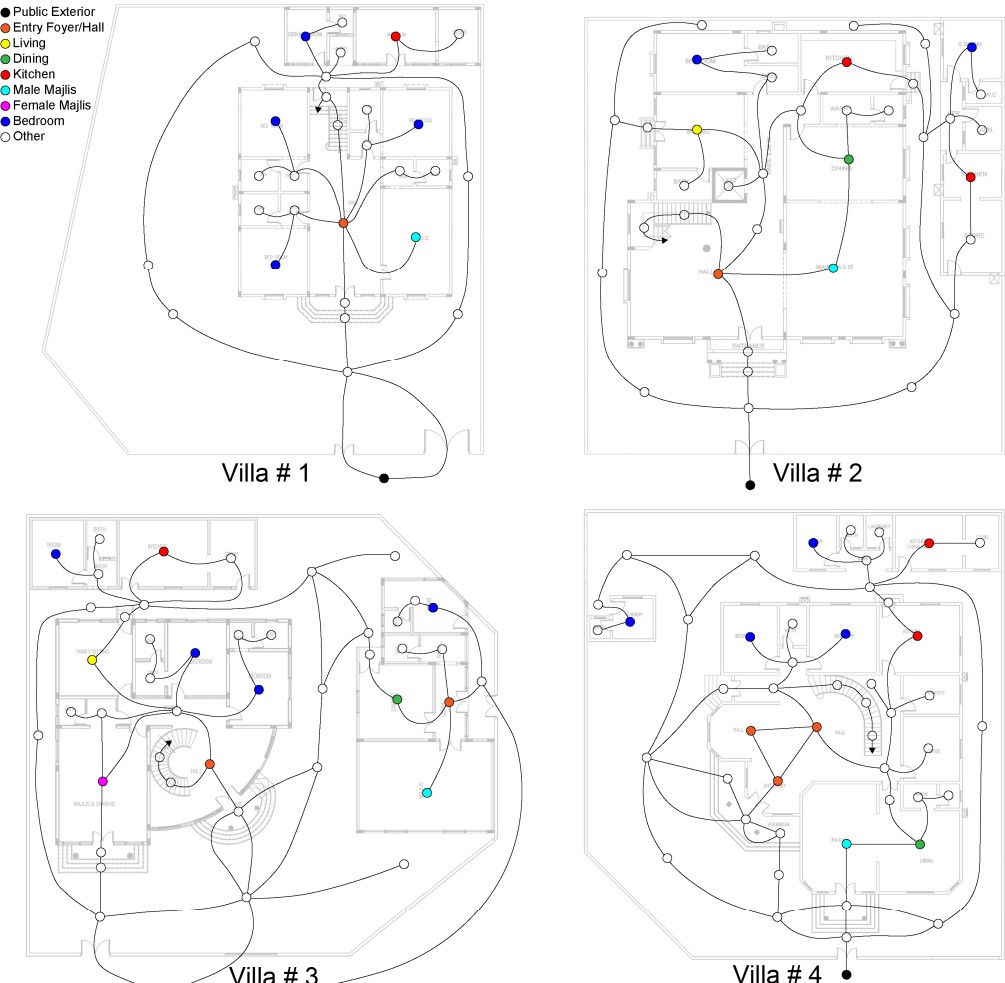

**Figure 4.** Relational graphs of convex spaces for the ground floor plans of the four contemporary villas with color-coding of key functional spaces (Source: Author).

The male majlis completely separates on an isolated branch of the graph in Villa #1, immediately adjacent to the entry hall of the main house, and Villa #3, which isolates the male hospitality spaces in a separate housing block with its own entry from the street. Villa #3 is the only one to possess a distinct female majlis space lying on a ring of circulation associated with a separate entry to the exterior space and the hallway providing the main distribution point at ground level in the house. The male majlis lays on a ring of circulation associated with the dining room in Villa #2 and #4, with the former immediately adjacent to the entry hall of the main house (such as Villa #1) and the latter possessing a separate entry, which is near to compound entry to the street in terms of metric distance. Villa #3 and #4 utilize an exterior transition sequence involving two additional convex spaces before entering the male majlis.

Two villas (#1 and #3) completely isolate the kitchen in a separate housing block. Access to the main house is via an exterior space, a sitting room in Villa #3, and a stairway

landing in Villa #1. The last, combined with the main entry stairs, indicates a marginal elevation rise from front to back on the residential plot of Villa #1. The other villas have a kitchen interior to the main house and another kitchen in the separate housing block associated with servants. Storage is adjacent to the kitchen in the separate housing block of all four villas. The interior kitchens of Villa #2 and #4 lay on a ring of circulation to the exterior space accessing the separate kitchen/servant blocks and via hallway transitions to the dining room. It indicates a division of duties for these kitchens, with storage and food preparation occurring with the separate kitchen/servant blocks and presentation and service associated with the interior ones. Unsurprisingly, every bedroom in every villa separates into an isolated branch of the graph, indicating their relative isolation for family members, guests, or servants. Finally, separating blocks in all four increases the importance of exterior yard spaces as transition spaces between distinct parts of the villa, especially the main house and the kitchen/servant block.

We included hospitality and service spaces within the definition of public space for these contemporary residential villas since these spaces most commonly involve non-members of the family, i.e., visitors and servants. The public space in the layout of Villa #1 consists of the ground floor, an intermediate floor landing, and an external service block. There are two entry points for people and vehicles on the southern compound fence. The lot coverage of this villa is about 35%. The villa's largest space is the main house's entry hall, measuring 10 m × 5 m. The male majlis is immediately adjacent to this entry hall, providing access to a small bathroom/washroom. The entry hall connects to the staircase providing a vertical transition to the first floor and intermediate landing, giving access to the exterior yard for transitions to the kitchen/servant block. In terms of private space, the entry hall provides access to three bedrooms with their own bathroom. In each case, bedroom and bathroom space access occurs via a vestibule transition space. Noticeably, the door swings from the entry hall to the bedroom vestibule and from the vestibule to the bathrooms invert compared to each other in a manner atypical in Western examples. The door swings into the bathrooms are purely functional in terms of maximizing the usable metric area for the room function. The door turns into the vestibule functions to obscure the largest metric area of the spaces for the bedrooms opposite the majlis. Interestingly, it does not necessarily limit visibility into the bedroom space itself. However, it does ensure if both doors are partially open that, the bathroom space remains completely opaque to the entry hall for three bedrooms. It suggests these are primarily guest bedrooms (especially the one adjacent to the main entry) with the potential for older male family members (son or grandfather) using the back bedrooms, dependent on the number and age of family members occupying the first floor of the main house. There is also a bedroom for a female domestic servant in the separate kitchen/servant blocks at the rear of the compound.

Regarding the visual integration overlay, the two locations with the most muscular visibility are the southwest corner of the exterior yard and the main house's entry hall (Figure 5). The first is due to the available metric area visible from this location. The second is due to the entry hall being the largest interior space and its importance as a control point for connection to the house's distinct parts. In terms of cultural influences detectable in the spatial layout, there is a marginal increase in visual integration for the hospitality space (i.e., male majlis) compared to the ground floor bedrooms due to its direct connection to the entry hall and the intermediate effect of the vestibule spaces to the bedroom suites. In terms of privacy and gender separation, the bedrooms for the main family members (especially any females) separate into sections via a vertical transition on the first floor. The ground floor bedrooms are marginally separate for visual integration from the public spaces (entry hall, stairway, and hospitality) due to the transition via the vestibules. Due to the importance of the entry hall within the main house's layout, this family needs to adopt a regimented routine for visitors to mitigate any effect on private domestic functions, especially for female visitors and family members. It suggests male family members vacate the villa at specific times of the day, whether for work, school, and socialization occurring

in public venues away from the house. It is also consistent with how Qatari households tended to function in traditional homes of the past.

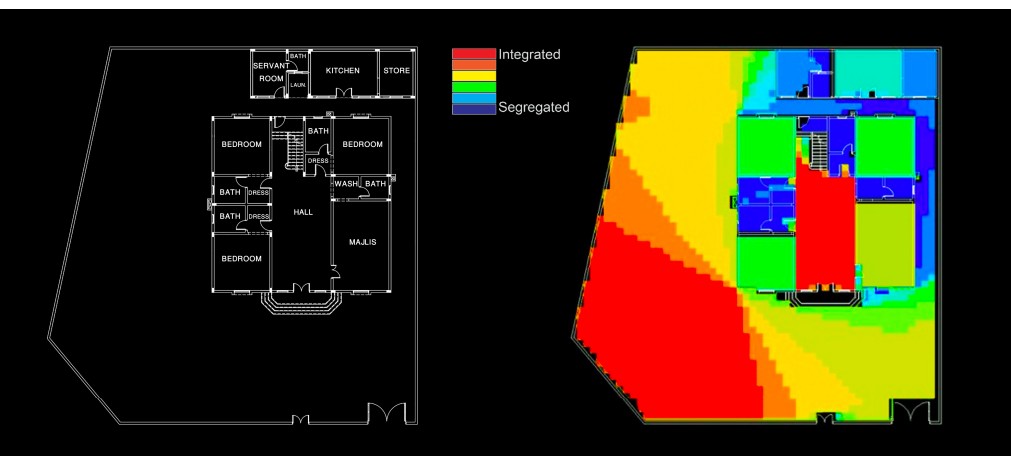

**Figure 5.** (**left**) Plan of ground floor level and (**right**) visibility graph analysis overlay for Villa #1 (Source: Authors).

In terms of the inhabitants' activities, the large entry hall serves a multi-functional role due to the absence of a formal living room (in Western terms) or dining space on the ground floor. Historically, room functions in traditional Qatari houses also tended to serve multi-functional roles [14,18]. In the case of Villa #1, the entry hall serves as both the greeting space for visitors before entering the majlis at times and the main living/dining area for the family. The entry hall's metric area is large enough to accommodate multiple furniture layouts, supporting its daily use for family gatherings and leisure. The connective importance of the space also suggests it is more than a mere entry hall. Similarly, the large, integrated exterior yard space is appropriate for outdoor activities and family gatherings during the milder months of the year in Qatar's climate, i.e., late November to March. For other times of the year, this exterior yard is large enough to accommodate multiple parked vehicles and three-point turns for vehicular drop-offs. Finally, there is only a narrow view from the street into this part of the exterior yard with a near 180° opening of the vehicular access gates, and there is no view at all if the gate only opens at a 90° angle.

The separate kitchen/servant block and the exterior yard spaces approaching it are the most segregated spaces in the residential compound. This segregation enables flexible movement via a visually isolated space for females of the household between the main house and separated kitchen block to coordinate servant activities and food preparation. It isolates the domestic servant within the home, especially from visitors. However, it also means the domestic servant's bedroom suite possesses a similar degree of privacy horizontally in the plan that the first-floor bedrooms for family members achieve vertically in the section on the first floor. It reflects all bedrooms operating on separate sequence branches of the relational graph, whether for family members, guests, or servants.

The public space in the layout of Villa #2 comprises the ground floor and the external service block on the eastern plotline (Figure 6). A single gate provides access from the street into the residential compound. The lot area coverage of the residential villa is about 55%. The principal entry to the main house directly faces the only gate on the southern compound wall. Like Villa #1, this villa possesses a large entry hall with the majlis immediately adjacent through an arcaded doorway to the left of the main entrance. Connected to the majlis is the dining room, and washrooms are easily accessible from the dining room. Even though the majlis/dining room suite lays on an internal ring of circulation, this layout effectively isolates the hospitality spaces by using transition spaces elsewhere in the plan. There is also a large sitting room with access to a private exterior side yard, which could be a living space for the immediate family. For private space, the main house's ground floor contains a single-bedroom suite with a bathroom at the rear

of the house accessed via a vestibule. We previously discussed the functioning of the interior kitchen related to the separate kitchen/servant block. The nearest to the kitchen and relative seclusion of the bedroom at the rear of the house suggests this could be an older female family member's bedroom, i.e., grandmother. Villa #2 is the only house in the sample with an elevator to access the first floor. There is also vertical access to the first floor via a stairway adjacent to the elevator on the entry hall's far wall opposite the main entrance. The first floor contains the family bedrooms. The most isolated ground-floor bedroom is for the domestic servant in the separate kitchen/servant block at the lot's northeastern corner. It represents the maximum allowable distance from an entry gate on the southern compound wall to the opposite corner of the plot without allowing potential visitors to intrude on the secluded family side yard to the west.

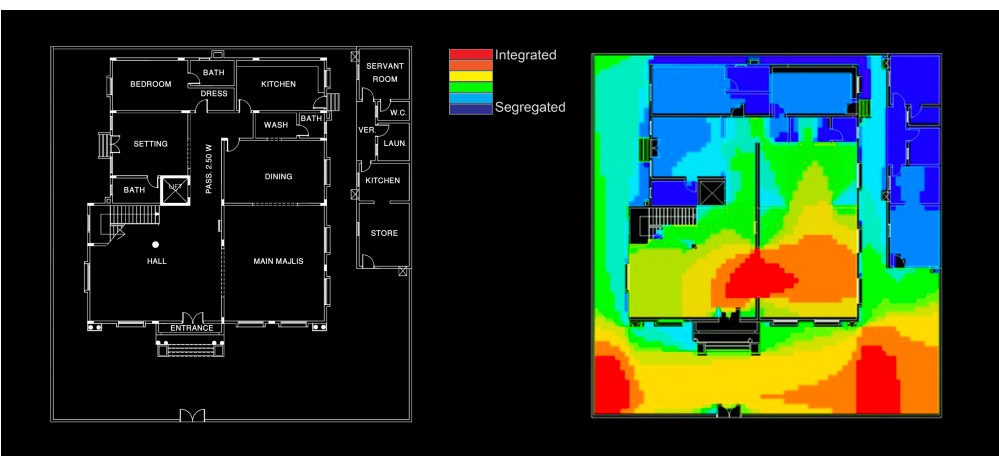

**Figure 6.** (**left**) Plan of ground floor level and (**right**) visibility graph analysis overlay for Villa #2 (Source: Authors).

Regarding the visual integration overlay, the two most integrated exterior locations are the intersection of the front and side yards. It is due to the front yard's metric area and the effectively dead-end nature of the side yards, which do not connect to form an external ring of circulation. The entry hall and majlis are the most integrated spaces for the internal layout. The majlis is more integrated due to the large opening to the dining room. Their combined metric area is about twice as large as the entry hall's main space, so integration skews towards the majlis. Despite the connective importance of the hallways to the rear operating as control spaces for distribution in the main house, there is only a moderate degree of visual integration for these spaces. The interior kitchen is the most segregated space in the main house, the rear bedroom, and the sitting room. This segregation occurs even though the sitting room and interior kitchen lay on circulation rings to the exterior yard. The sitting room's segregated nature also indicates this space might serve female visitors, who can enter via the side door if male visitors use the main majlis and enter via the entrance hall. This feature re-interprets a similar approach in traditional Qatari courtyard houses. Such homes would possess a side entry for the family and female visitors' access to maintain visual and functional privacy [70–72].

Despite the larger metric area of the exterior front yard at the south-eastern corner of the lot, the side yard exterior space between the main house and separate kitchen/servant is more segregated than the family side yard on the opposite side. It generates a similar visual relationship between the main house and kitchen/servant block as Villa #1, indicating flexible movement for female household members. For the hospitality spaces, there is a broader range of visual integration from highly integrated (in red) at the threshold between the entry hall and majlis and the effective useable area of the majlis (in orange) to moderate integration in three corners (in green) of the room. It indicates a sophisticated degree of spatial differentiation within the majlis itself based on where anyone sits or stands. This visual integration pattern accurately reflects the majlis's importance for hospitality in

contemporary Qatari families' social life. For the activities of inhabitants, the entry hall, majlis, and dining room is the primary setting for the largest gatherings of the family. The immediate family probably utilizes the sitting room or dining room for everyday activities. The sitting room possesses an informal ambiance due to its segregated nature and direct access to the exterior side yard, facilitating privacy for family activities.

The public space of Villa #3 includes the ground and two separate blocks for an external kitchen/servant quarters on the northern lot line and hospitality on the eastern lot line (Figure 7). The hospitality block has separate access to the street and a side entrance connecting the dining room of the majlis to the yard. Principal access to the residential compound is via the main gate and a larger sliding gate for vehicles to the south. There is a second majlis with its own entrance for the use of females within the main house. Villa #3 is the only one with a female majlis in the sample. The separate access to the female majlis aligns with the main footpath gate. In contrast, the entrance to the main house's entry hall is closer to the vehicular gate, enabling vehicular drop-offs for family members. Access to the separate kitchen/servant block occurs via a sitting room. An articulated vestibule is a transition space between the sitting room/internal hallway and the female majlis, providing additional female gatherings in the hospitality space.

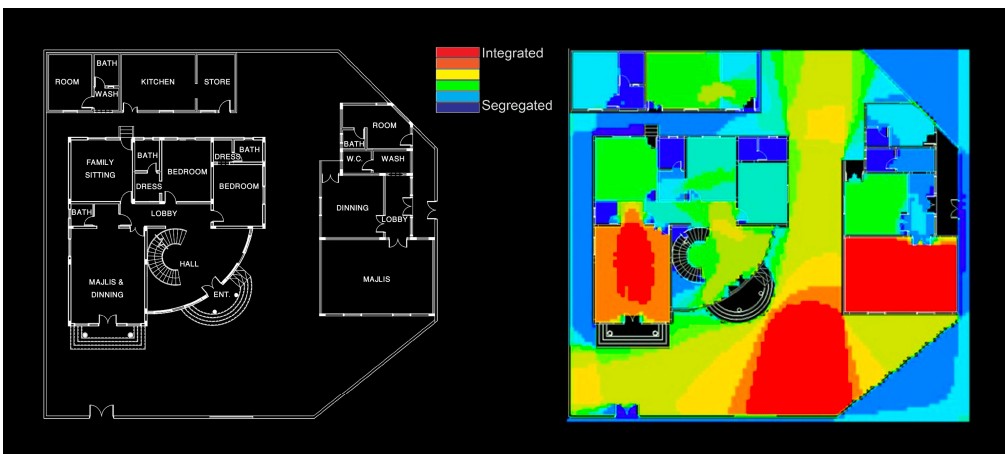

**Figure 7.** (**left**) Plan of ground floor level and (**right**) visibility graph analysis overlay for Villa #3 (Source: Authors).

The private space of the villa consists of the first floor and penthouse space. On the ground floor, two en-suite bedrooms have their doorways away from the vertical access via the stairway to the first floor. The two bedrooms could function for older family members or guests. The central bedroom is more for grandparents and the eastern one for guests or a more senior male family member, such as a grown son. There is a bedroom in the hospitality block for use by a male domestic servant. There is also a bedroom suite for a female domestic servant in the separate kitchen/servant blocks at the rear of the plot. It is the maximum metric distance feasible away from the street entries on the southern and eastern compound walls.

For the visual integration overlay, we had to model a narrow strip of the public street outside the compound due to the separate entries on the southern and eastern perimeters. There is a high degree of visual integration for the exterior yard near the principal entry to the main house and vehicular access gate, primarily due to a sizeable metric area in this location. It is the vehicular drop-off area for the family. Both male and female majlis possess high visual integration due to their large metric area and strategic location in the main house and hospitality block, respectively. There is only a moderate degree of visual integration in the entry hall of the main house. There is a sharp contrast in integration levels between hospitality spaces and family-use spaces in the villa. Only the sitting room of the main house, the kitchen of the separate rear block, and the hospitality block's dining room possess moderate levels of integration due to their location on rings of circulation

with the exterior yards. The bathroom spaces of the bedrooms are the most segregated spaces in the villa. However, the layout also secludes the bedrooms themselves.

The male and female majlis' hospitality spaces are the largest ones in Villa #3, with implications for gender segregation. The majlis in the main house is primarily a female space with a separate entrance near the compound's main gate. Female visitors can arrive without encountering any male members of the household. There is also an internal vestibule to this majlis space, which marginally separates it from the house's sitting room and rest. The majlis in the separate hospitality block is a male space separating from the main house. Male visitors and relatives can readily access the hospitality block without disturbing any household functions or encountering other household members. The hospitality block also possesses a dining room where an entry foyer space mediates the residential compound's internal access point. Male and female domestic servants are separated into bedroom suites available in the hospitality and kitchen blocks. A male domestic servant occupies the hospitality block's bedroom suite, while a female domestic servant uses the bedroom within the kitchen block. This layout helps to ensure gender segregation within the acceptable limits between each while allowing ease in performing their household duties. The settings for the public activities of inhabitants are the majlis areas. Providing two majlis spaces enables flexibility for the scale and choices for gatherings and events. The entire interior space of the main house is the setting for the inhabitants' private activities, except for the female majlis. Family members can utilize the sitting room for leisure or meals (due to its proximity to the separate kitchen/servant block). The central hallway corridor is the main connective point to control distribution on the main house's ground floor, except for access to the first floor occurring via the entry hall. The entry hall is large enough to accommodate various furniture layouts, further impeding functional or visual access to the household from the front door, if necessary.

The public space of Villa #4 comprises the ground floor and a separate kitchen/servant block to the rear of the lot (Figure 8). There are three entrances: one to the majlis space aligned to the gate access to the street and the main entrance to a large entry hall convexly subdivided into three distinct spaces functioning for different purposes, i.e., foyer for greeting, adjacent sitting area for occupation, and a transition hall primarily for movement. There are two secondary entrances. A hallway space adjacent to the entry hall to the side yard serves as a distribution point to one side of the main house. A rear entrance connects the main house's interior kitchen and a separate kitchen/servant block. The private space includes the first floor and a penthouse. Two ground-floor bedrooms are accessed from a vestibule off the hallway providing access to the side yard. This bedroom suite with a shared bathroom on the ground floor operates as a guest wing, visually and spatially separated from the rest of the house. The main house itself and metric distance from the gate accessing the street serve to isolate the rear kitchen/servant block with the domestic servant bedroom visually and spatially within the layout. The same occurs with a small housing block for the gardener by orienting its exterior vestibule space to the rear of the plot.

Villa #4 possesses the most evident "front-back" lot pattern for the exterior spaces regarding the visual integration overlay. It arises due to the smaller gardener housing block shaping the spatial nature of a large rear yard. Like Villa #3, there is a large integrated area to the front of the exterior yard for vehicular drop-off adjacent to the main house's entry portico. Internally, the spaces with the highest visual integration are the main house's entry hall and the hospitality wing (majlis and dining room). The bedroom suite on one side of the main house's ground floor and the office on the other are the most segregated spaces of the main house. The interior kitchen possesses slightly more visual integration due to a ring of circulation outside to the rear kitchen/servant block and visibility down the eastern hallway of the main house to a vestibule accessing the hospitality wing. The villa's most segregated spaces are the domestic servant bedrooms/baths and the laundry in the separate rear blocks.

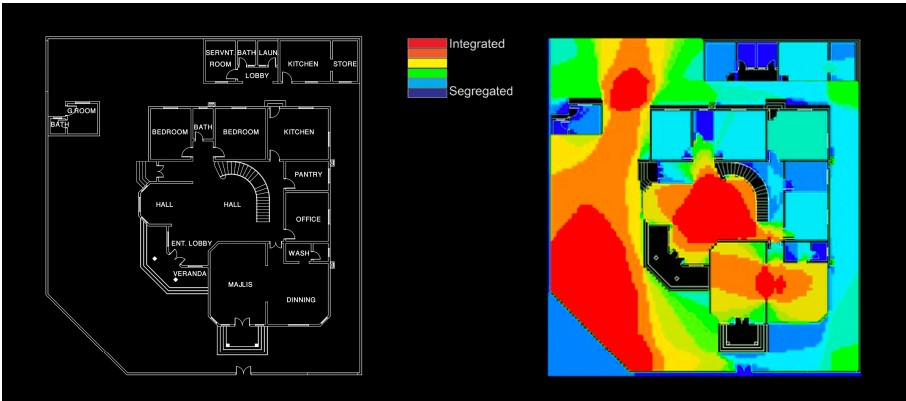

**Figure 8.** (**left**) Plan of ground floor level and (**right**) visibility graph analysis overlay for Villa #4 (Source: Authors).

The separation of hospitality spaces enables controlling access and use by male visitors. It could also work as a flexible space for female usage at other times, if necessary. The entry hall is also large and flexible enough to accommodate female visitors, giving them a priority for accessing the main house's interior spaces due to the hierarchal construction of privacy in Muslim homes [73,74]. Both spaces are suitable for the inhabitants' public activities, depending on the gender and nature of the event. The highly integrated entry hall operates like an interior climate-controlled courtyard separating parts of the ground floor plan, such as traditional Qatari courtyard houses [4]. The spaces of separate house branches, mediated by the entry hall, serve as the setting for the household's private activities, such as sleeping, dining, working, and reading, which require more privacy and spatial seclusion. The stairway's layout and position also function as a screen device to limit the visibility of the kitchen and pantry spaces.

## 5. Discussion

The sample's contemporary residential villas possess a complex arrangement relating the lot and the street, interior and exterior spaces, and different housing blocks to each other. It occurs even though the house form and spatial configuration follow a standardized approach from common construction practice. The analysis results tend to invalidate any hypothesis about the loss of socio-cultural roles in the spatial form of contemporary Qatari houses. Quite the opposite, they are physical, spatial, and configurationally designed to reflect the essence of privacy, gendered spaces, and the hospitality culture in contemporary Qatari households.

If we qualitatively examine the house genotypes ("general type") for each villa, based on a comparison of the pattern of the visual integration overlays for all four villas and focusing on the most important space and room functions (Figure 9), we find:

- Villa #1: Front yard = Entry Hall > Majlis > Gate Entry > Interior Kitchen = Bedrooms > Rear Yard = Kitchen/Servant Block;
- Villa #2: Front yard > Majlis > Entry Hall > Gate Entry > Dining Room > Bedrooms = Interior Kitchen = Side Yard > Kitchen/Servant Block;
- Villa #3: Front Yard > Male Majlis > Female Majlis > Gate Entry > Entry Hall > Entry Gate > Dining Room > Kitchen/Servant Block > Rear Yard > Bedrooms;
- Villa #4: Front yard = Entry Hall > Majlis > Rear Yard > Interior Kitchen > Entry Gate > Bedrooms > Kitchen/Servant Blocks.

Generally, using numerical weighing, this translates into a contemporary Qatari housing genotype of:

- Front Yard > Entry Hall > Majlis > Gate Entry > Interior Kitchen > Dining Room > Rear/Side Yard = Bedrooms > Kitchen/Servant Block.

The first four (front yard, entry hall, majlis, gate entry) strongly relate to public space access and hospitality functions. This genotype's next most integrated spaces (interior kitchen and dining room) relate to food preparation. The main house bedrooms and rear/side yard access tend to share similar degrees of spatial seclusion for distinct reasons: household privacy for the former and service management for the latter. Finally, the kitchen/servant block tends toward the most spatial and metric segregation in generating a household hierarchy between family members served and domestic employees providing the service.

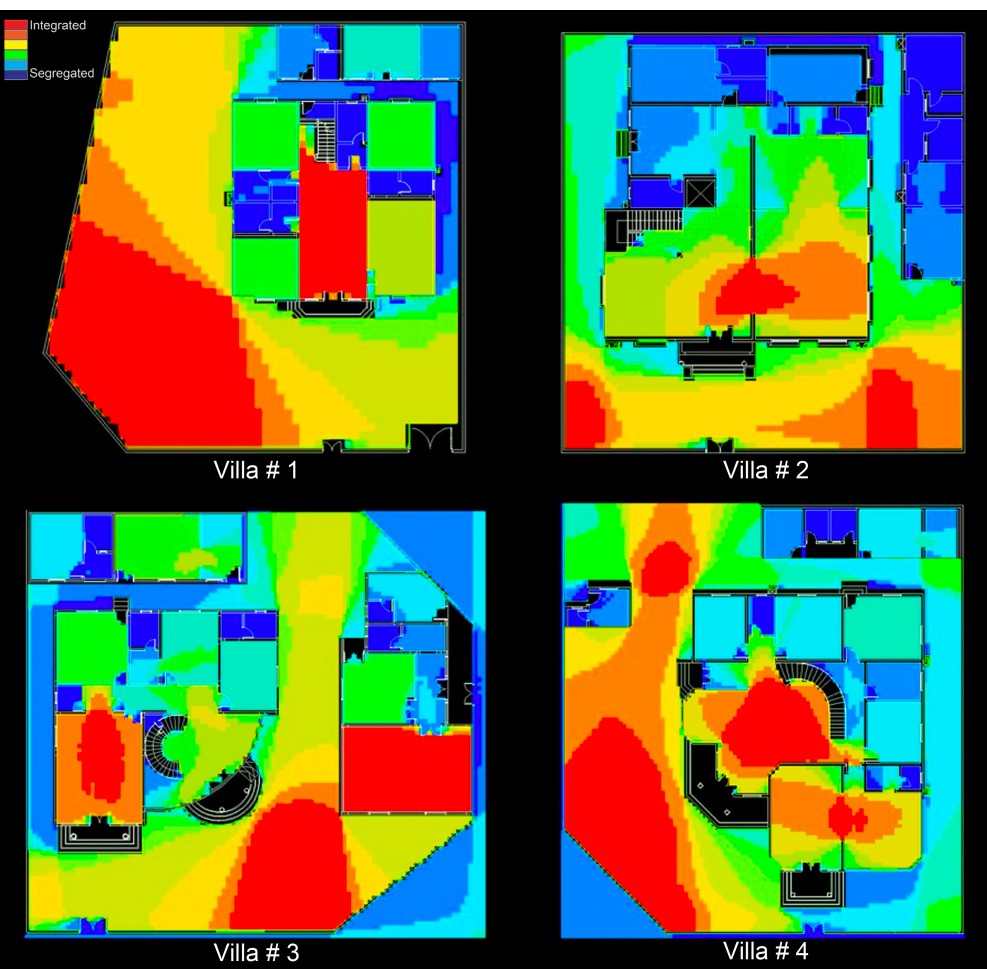

**Figure 9.** Comparison of VGA overlay for the ground floor plans of the four contemporary villas (Source: Authors).

An additional layer of household privacy mediates the visitor experience due to the entry gate(s) placement in moderately integrated locations on outer compound walls. Visitors do not enter the most integrated exterior yard space, thereby gaining an "all-at-once" visual understanding of the compound layout. However, visitors also do not enter more segregated locations, which might confuse them. Instead, the entry points for hospitality spaces are immediately adjacent and directly visible from the entry gate. The visitor experience is a carefully controlled sequence whereby they cross the entry gate's threshold, see but do not access the integrated front yard, and enter the majlis directly or via an entry hall. It provides contemporary Qatari residential villas with relatively straightforward "front of the house" (entry/hospitality) and "back of the house" (service/bedrooms) realms. Visitors are aware of it but generally do not experience it. There is an inverted experience of the household for domestic servants. Domestic servants know the home's entry/hospitality realm but only tend to experience it after the conclusion of

events or gatherings. A well-defined series of spaces mediate between these two realms in the household for the interaction of family members and domestic servants. Finally, everyone, whether family members, guests, or domestic servants, enjoys a high degree of privacy in their bedroom suite spaces. Collectively, it arises from the design of the spatial layout, careful patterning of visibility and access, and the utilization of separate housing blocks in these contemporary residential villas.

We approached this study with three key criteria: (1) the spatial form of the villa and (2) cultural roles, and (3) the activities of the household inhabitants. The study's findings indicate the usefulness of this approach for understanding the spatial culture of house form based on the socio-cultural imperatives of society. Such an approach has helped us know vernacular settings and traditional housing models. In this regard, we encourage readers to refer to the available literature about traditional courtyard houses in the Middle East and the Arabian Gulf's geographical context to further explore the architecture of the courtyard house as a spatial and cultural model [75–78].

Whether these contemporary residential villas provide an ideal model for Qatari households today is open to debate. It depends on the family's socio-spatial priorities. Villa #1 offers ample exterior yard space for outdoor household activities or the parking of vehicles. Its allocation of outdoor yard space is more typical of historic Qatari houses, though this occurs around the main house instead of inward-looking on a courtyard. Villa #1 accomplishes this by emphasizing the importance of the large entry hall for connections and the distribution of public and private household activities, thereby increasing the probability of chance encounters or visibility by guests in the villa. A less socially conservative Qatari family would find this spatial layout satisfactory for their needs. However, it could be problematic for a more conservative one. The spatial arrangements of Villa #2, #3, and #4 all achieve a degree of balance between the public and private activities of the household, which could prove sufficient for most Qatari families, socially conservative or otherwise. All three do so by adopting different spatial strategies, which we could define as (1) side yard articulation in Villa #2, (2) hospitality separation in Villa #3, and (3) interior hall articulation in Villa #4.

However, Villa #2 accomplishes its sophisticated articulation of the side yard spaces at the expense of adequate outdoor yard space for private family gatherings. The size of its private side yard seems too small for outdoor family activities, and additional outdoor space is well-spent at the plot's rear. Villa #3 balances the family's public and private activities at the expense of a moderately side rear yard space, which seems inappropriate for private family gatherings and more characteristic of a utility yard, especially for domesticated animals. Its large front yard space could serve outdoor private family activities if the compound walls are high enough to screen such gatherings from two public streets effectively. Finally, Villa #3 achieves a similar balance between the family's public and private activities as Villa #2, with its large rear yard space bearing the characteristics of a utility yard due to the small servant block's adjacent location. However, the side yard space of Villa #3 is large enough to accommodate private family outdoor activities, unlike Villa #2. The degree of functional and gender separation in Villa #3 is effective for the most socially conservative Qatari families. However, less conservative ones might find the well-articulated spatial separation of Villa #3 a little too rigidly regimented for everyday use.

## 6. Implications of the Study

The study has important research outputs about architectural and urban identity in domestic residential architecture. It can help to resolve some current challenges for Qatari housing. There is an opportunity for application to the more prominent Arab Gulf and GCC region. It includes but is not limited to the inharmonious built form and spatial layout issues. The study offers a structured methodology using space syntax to assess and analyze contemporary housing schemes in a well-defined socio-cultural setting. The analytical outcomes provide valuable insights for addressing architectural and design deficiencies

and inefficiencies. This represents a considerable contribution regarding the applicability and convenience regionally and internationally.

In the research design section, the authors discussed the rationale for selecting the unit of analysis, namely the four contemporary villas, to clarify the selection criteria and the sampling method. Moreover, in the study's findings, the authors interpreted the data about the methodological framework based on the requirements for analysis, including the spatial form, socio-cultural factors, and activity system in the four villas. In this way, the meanings of the research results are interpretative, fitting the purpose of the study to qualitatively evaluate issues of the spatial culture of housing in Qatar.

We recommend further studies involving a larger sample size and VGA modeling of the upper floor levels to provide additional insights about the layering of privacy vertically in sections in contemporary housing models. We anticipate that the analytical outputs of modeling other floors will prove highly segregated in relative terms compared to the ground floor level. We also expect that such models might need more spatial differentiation to evaluate subtle differences between public and public space in these house models. The ground floor of the main house would become more highly integrated overall as the primary spatial layout connecting the upper floors and separate housing blocks. On the other hand, modeling upper floors might enable architects and designers to reinvigorate rooftop spaces as overlooked or forgotten social spaces for the private use of families in contemporary housing schemes of the region.

Finally, a more extensive study considering the neighborhood scale of residential areas related to the spatial culture of housing could enrich further research into architectural sociology and the communal value of the house as a social unit of space in a broader setting. We anticipate that contemporary planning and zoning regulations will negatively impact the socio-cultural ties between family households and the larger community by under-utilizing opportunities for social interaction outside the house in public urban space. It was an essential attribute of neighborhood formation in vernacular schemes and individual obligations within the unifying system of Islam societies.

## 7. Limitations

In the relevant seminal literature and within the scope of space syntax studies, there are multiple spatial and socio-spatial analysis methods ranging from visual analysis to mathematical and statistical analysis, with the approach combining these techniques [19,79,80]. The authors addressed such methodological concerns explicitly in the literature review and the research design sections with an apparent reference to the multiplicity of the techniques used in the spatial and socio-spatial analysis of housing. However, the research study focuses on visibility and modeling visibility to interpret the probable relation to socio-cultural factors such as gender roles, hospitality, and privacy. The research study aligns with its focused objectives to provide consistent results. As an advancement of the current study, a combined methodology would enrich the outcomes and further explore the spatial culture in housing in Qatar.

Modeling is another limitation of the study as the visibility graph analysis (VGA) only occurs for the contemporary residential villas' ground floor plans. The four villas schematically extend vertically in sections to include first-floor and penthouse spaces, which might have further implications for the study's findings. However, as argued in the paper, allocating public and private space on the ground floor in the main house and separate housing blocks is a common trend in contemporary Qatari residential villas. It proved sufficient to provide a comprehensive picture of functional spaces encompassing privacy, gender segregation, and hospitality issues in the case studies. The small number of case studies (4) in the sample is also a limitation due to time and resource constraints.

## 8. Conclusions

This paper's methodology produced useful analysis for interrelating socio-cultural patterns to domestic spatial forms in research outputs. It requires researchers to revisit

the objectives and hypothesis of such a study to verify the results. This study explored architectural and urban identity in domestic house forms in contemporary Qatari residential villas. The study fills a gap in our knowledge between architecture and sociology by enriching interdisciplinary research linking two aspects of the built environment. First, the house form and culture of the current widely spread housing model of the free-standing, suburban residential villa. Second, the socio-cultural attributes of contemporary Qatari society itself. Therefore, the study helps to shed further light on current trends in housing, driving the urban transformation of rapidly growing Middle Eastern cities such as Doha. The study included qualitative evaluation as well as modeling simulation using space syntax. It allowed us to assess more deeply the issues of architectural configuration, spatial distribution and location, and the relationship of both to inhabitant's systems of activities and practiced culture such as hospitality. It helps to ensure that research outcomes are objective, holistic, and reasonable in providing architects, designers, and policymakers with better tools to resolve contemporary housing challenges, especially in the GCC region.

Based on this study, we argue for the relevancy of the following design initiatives in contemporary Qatari housing. Firstly, the cultural pattern of privacy should emerge, almost naturally, through the spatiality of the architectural object of the residential villa itself based on well-designed and thoughtful patterning of spatial configurations without sacrificing cohesive architectural forms or compact residential lots.

Secondly, gender spatial segregation in households requires thoughtful practice in design, especially related to the manners, ethics, and social value of women in conservative Islamic societies' domestic settings. Spatial segregation does not necessarily equate to social inequality. It can equate to a social role. Societies may still highly value such social functions, regardless of their specific articulations with the residential spatial layout. It requires the sensitivity of architectural form to the inhabitants' social needs while allowing males and females equitable access to spaces within the private realms of the household. Such design objectives tend to require spatial sophistication, not formal simplification. For example, traditional Qatari courtyard houses tend towards formal simplification but possess a deep spatial complexity in their vernacular design. These contemporary Qatari residential villas had evidence of a similar approach to house form and culture.

Lastly, hospitality is a defining feature of Arab culture. It shapes the spatiality of contemporary housing schemes, and limited contemporary lot sizes devote a notable amount of metric area to hospitality. Better utilization of hospitality areas in contemporary residential villas should be an aim for design, function, timing, and usability for different household constituencies. It suggests a need for greater adaptability in design for the system of household activities associated with majlis spaces to ensure its full, not occasional, sustainable utilization beyond rare circumstances.

**Author Contributions:** Conceptualization, A.A.-M., R.F. and M.D.M.; methodology, A.A.-M. and M.D.M.; software, A.A.-M. and M.D.M.; validation, A.A.-M., M.D.M., R.F. and R.S.A.-M.; formal analysis, A.A.-M. and M.D.M.; investigation, A.A.-M., M.D.M. and R.F.; resources, A.A.-M., M.D.M. and R.F.; data curation, A.A.-M., M.D.M. and R.F.; writing—original draft preparation, A.A.-M., M.D.M. and R.F.; writing—review and editing, A.A.-M., M.D.M., R.J.I. and R.F.; visualization, A.A.-M. and M.D.M.; supervision, R.F. and R.S.A.-M.; project administration, R.F. and R.S.A.-M.; funding acquisition, R.F., R.S.A.-M. and R.J.I. All authors have read and agreed to the published version of the manuscript.

**Funding:** This research received no external funding.

**Data Availability Statement:** No new data was created or analyzed in this study. Data sharing does not apply to this article.

**Conflicts of Interest:** The authors declare no conflict of interest.

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
