# Peer review of "Investigation of Spatial and Cultural Features in Contemporary Qatari Housing"

_urbansci, doi:10.3390/urbansci7020060_

Round 1

Reviewer 1 Report

The paper entitled "The Spatial Culture of Gender, Privacy, and Hospitality in Qatari Contemporary Housing Using Space Syntax” falls within the scope of the Urban Science Journal and shows some technical relevance.

The manuscript provides interesting information and assess the social and cultural impact of architecture globalization in the spatial layout of four contemporary Qatari residential villas by means of space syntax tool to demonstrate patterns of visibility and room relations in the samples to understand users´ activities in the domestic setting.

The paper includes an adequate methodology and results support the conclusions of the study. Therefore, the work could be publishable in its current state.

Author Response

Dear Reviewer,

Thank you very much for the valuable comments and positive feedback.

We appreciate it.

Rima

Reviewer 2 Report

Thanks for giving me the opportunity to review research paper entitled "he Spatial Culture of Gender, Privacy, and Hospitality in Qatari Contemporary Housing Using Space Syntax".

Overall, the manuscript is original and adds to knowledge and practice. The authors have done a good job in analyzing the four cases and concluding final remarks about the role of contemporary housing in social and cultural aspects. The manuscript could benefit from the following suggestion:

-        I think the current title does not reflect the paper. Please re-think about revising the title

-        Abstract needs restructure. It should start with the purpose of the research then the methodology, key findings and finally major implication.

-        The purpose of the paper needs re-write. For examples lines 44-45 “The paper utilizes space syntax to assess and evaluate the influence of socio-cultural factors in the modern house form”.  I think you cannot examine the influence without statistical data. You could examine the role of …….. in ……. But not the influence/effect or impact. You may need to revise this throughout the manuscript.

-        Research hypotheses in lines 55-57  should be based on theoretical foundation and hence it should come after the background please with more discussion and supporting arguments

-        Howe the qualitative data was collected, analyzed and how did you ensured data validity and reliability. This should be discussed in the methodology section.

-        Please split revise limitation and practical implication section  

Author Response

Dear Reviewer,

Thank you very much for your valuable comments and suggestions.

Kindly find attached our feedback as per your recommendations.

Thanks

Rima

Reviewer 3 Report

Presented article with Title ” The Spatial Culture of Gender, Privacy, and Hospitality in Qatari Contemporary Housing Using Space Syntax” is writing on 28 pages with 11 figures and 80 references. This article is written clearly and comprehensibly.The article focuses on a practical example with a specific data.

Suggestions:

-          From article are not clear the results and added value.

-          In article lacks more technical specifications.

-          Conclusions require redrafting. They are too short and inconsistent.

-          Conclusions, please describe your future directions.

After review I recommend publish this article after minor revision in journal.

Author Response

(The authors gave the same response as above.)

Round 2

Reviewer 2 Report

Thanks for considering most of my comments on the earlier version. However, i still did not see any addition in relation to qualitative data collection and analysis. I suggest you make conclusion as the last section please. Please merge contribution and practical implication sections as one section under the title  "implications of the study". best wishes  

Author Response

Dear Reviewer,

Thank you very much again for your review and valuable feedback.

Kindly find attached our response and the updated manuscript.

Thanks
